# Floral Homeotic Factors: A Question of Specificity

**DOI:** 10.3390/plants12051128

**Published:** 2023-03-02

**Authors:** Kevin Goslin, Andrea Finocchio, Frank Wellmer

**Affiliations:** Smurfit Institute of Genetics, Trinity College Dublin, D02 PN40 Dublin, Ireland

**Keywords:** flower development, organ specification, Arabidopsis, MADS-domain protein, transcription factor specificity

## Abstract

MADS-domain transcription factors are involved in the control of a multitude of processes in eukaryotes, and in plants, they play particularly important roles during reproductive development. Among the members of this large family of regulatory proteins are the floral organ identity factors, which specify the identities of the different types of floral organs in a combinatorial manner. Much has been learned over the past three decades about the function of these master regulators. For example, it has been shown that they have similar DNA-binding activities and that their genome-wide binding patterns exhibit large overlaps. At the same time, it appears that only a minority of binding events lead to changes in gene expression and that the different floral organ identity factors have distinct sets of target genes. Thus, binding of these transcription factors to the promoters of target genes alone may not be sufficient for their regulation. How these master regulators achieve specificity in a developmental context is currently not well understood. Here, we review what is known about their activities and highlight open questions that need to be addressed to gain more detailed insights into the molecular mechanisms underlying their functions. We discuss evidence for the involvement of cofactors as well as the results from studies on transcription factors in animals that may be instructive for a better understanding of how the floral organ identity factors achieve regulatory specificity.

## 1. Introduction

Flowers contain the reproductive organs of angiosperms, the largest group of land plants. They produce much of the food humans and their livestock consume and are therefore pivotal for agriculture and the economy. Early genetic analyses of flower development led to the formulation of the ABC model of floral organ identity specification [1,2]. This model posits that a small number of floral homeotic genes act in a combinatorial manner to specify the four different types of floral organs (i.e., sepals, petals, stamens, and carpels). Specifically, according to the ABC model, the formation of sepals is controlled by A-class genes, petal development by the combined activities of A- and B-class genes; stamen formation by B- and C-class genes, and carpel development by C-class gene activity alone. Over the past 30 years, the central tenets of this model have been confirmed in diverse angiosperms. However, the model has also been expanded to incorporate D-class genes, which are involved in specifying ovule identity, and E-class genes, which are required for the activities of the A-, B-, C-, and D-class genes [2,3] (Figure 1).

Molecular cloning of the floral homeotic genes showed that most of them encode members of the family of MADS-domain transcription factors (named after four of the first family members identified, namely MCM1 from the yeast *Saccharomyces cerevisiae*, AGAMOUS from the plant *Arabidopsis thaliana*, DEFICIENS from the plant *Antirrhinum majus*, and SERUM RESPONSE FACTOR from the human *Homo sapiens*). The MADS-domain protein family is much enlarged in land plants relative to other eukaryotes (with more than 100 members in Arabidopsis [4]) and comprises many key regulators of reproductive development involved not only in floral organ specification and development but also in processes such as flowering time control and fruit formation [5].

Transgenic and mutant studies have shown that the transcription factors encoded by the floral homeotic genes are necessary and sufficient to define the identity of lateral organs as specific floral organs (summarized in [6,7]). Furthermore, gene perturbation experiments showed that these floral organ identity factors not only mediate floral organ specification at the earliest stages of flower development but are required throughout most of floral morphogenesis to control organ growth and differentiation [8,9,10,11,12,13]. This finding is in agreement with the prolonged expression of the corresponding genes during flower development [14]. Though initially thought to act as homodimers or as heterodimers in specific combinations [15], it was later predicted and shown that the floral organ identity factors can form tetramers to give rise to different regulatory complexes that mediate the specification of the different floral organ types [16,17,18,19] (Figure 1). It is currently unknown whether the formation of tetramers is necessary for floral organ identity factor function or whether dimers are sufficient for the regulation of at least some target genes. However, it has been shown recently that in Arabidopsis, tetramerization of the C-class factor AGAMOUS (AG) and the E-class factor SEPALLATA3 (SEP3) is required for the control of floral meristem determinacy [20], suggesting that tetramers of floral organ identity factors may carry out most, if not all, of their functions. The molecular mechanism(s) underlying the activities of these complexes are, however, still not well understood. Here, we review what is known about how floral organ identity factors control flower development. We focus especially on the question of how these closely related transcription factors achieve regulatory specificity. We argue that this specificity is most likely conferred by a combination of different mechanisms, similar to what has been suggested for other important developmental regulators such as Hox proteins in animals.

## 2. DNA-Binding Preferences as a Determinant of Specificity

It has been shown that homo- and heterodimers of MADS-domain transcription factors bind to so-called CArG-box sequences (consensus: 5′-CC-(A/T)_6_-GG-3′) and that their preferences for and affinities to individual CArG-box motifs vary depending on the composition of a given dimer [21]. The sequences flanking the actual CArG-box also appear to play a role in determining the DNA-binding specificities of these transcription factors both in vitro and in vivo [22,23,24]. Because there are many MADS-domain transcription factors encoded in plant genomes (see above), which can often interact if co-expressed, there is likely a vast array of dimers with different DNA-binding preferences, providing a rich pool for regulatory diversity.

Chromatin immunoprecipitation assays coupled to next-generation sequencing (ChIP-seq) revealed that several thousand sites are bound by each of the floral organ identity factors in the Arabidopsis genome [9,10,25,26,27,28]. As expected, the bound regions often contain CArG-box sequences, but there are also sites where no such sequences can be identified, suggesting that in some cases, the floral organ identity factors bind to other motifs. In agreement with the idea that these transcription factors can act together in complexes, large overlaps between their genome-wide DNA binding patterns were detected [10]. However, surprisingly, this is also true for the A-class regulators APETALA1 (AP1) and AG, which are not expressed in the same cells [29,30,31] and have not been recovered as part of the same protein complex in proteomics experiments [19]. Thus, it appears that these floral organ identity factors can bind to many of the same sites in the Arabidopsis genome in spite of their clearly distinct functions in flower development. This finding was confirmed and further expanded on in a study comparing ChIP-seq datasets for eight MADS-domain proteins, including the floral organ identity factors but also several regulators of flowering time [22]. It was shown that the binding patterns of these MADS-domain transcription factors show considerable overlap but that there are also clear differences, with some sites being targeted by only one of the factors analyzed. Based on these observations, it can be asked whether, or to what extent, the different functions of the floral organ identity factors (and, by extension, those of other MADS-domain proteins) are determined by their inherent DNA-binding activities.

In classic experiments, in which the amino-terminal half of the MADS-domain of different floral organ identity factors was replaced by the corresponding sequences of human MADS-domain proteins, it was shown that the expression of the resulting chimeric proteins led to gain-of-function phenotypes similar to those observed in lines overexpressing the unmodified floral organ identity factors [32]. Because it was observed that the DNA-binding preferences of the chimeric proteins were altered in vitro when compared to the unmodified transcription factors, it was concluded that the functions of the floral organ identity factors may not depend primarily on their precise DNA-binding specificities. The results of recent studies have challenged this view. For example, a comparison of data from high-throughput in vitro DNA-binding assays and genome-wide localization data showed that different floral organ identity factor complexes exhibit distinct DNA-binding preferences [23]. Furthermore, these preferences could be linked to specific sets of targets and developmental programs during flower development, suggesting that differences in DNA-binding preferences account for at least some of the specific functions carried out by different floral organ identity factor complexes. What could the molecular basis of this proposed mechanism be? It was shown recently that the so-called Intervening (I) domain, which follows the amino-terminal DNA-binding MADS-domain, is involved in determining the DNA-binding specificities and dimerization properties of floral organ identity factors and other plant MADS-domain proteins [33]. Notably, a domain swap experiment showed that the I domain of AP1 is sufficient to confer AP1-like activity on AG. It was therefore suggested that the I domain confers, to a large extent, functional identity to the floral organ identity factors. Expanding this work to other pairs of MADS-domain proteins should reveal whether the I domain is a universal specificity determinant of the plant MADS-domain protein family.

In summary, there is evidence that differences in the DNA-binding preferences of individual MADS-domain transcription factors (or MADS-domain protein dimers) account, at least in part, for their functional specificities. At the same time, the closely-related floral organ identity factors show considerable overlaps in their genome-wide binding patterns, raising the question of whether their DNA-binding activities alone are sufficient to determine their clearly separable functions.

## 3. Cofactors as Determinants for Specificity

To understand whether all the binding events that were detected by ChIP-seq were functional, transcriptomics experiments were carried out after altering floral homeotic gene activities. These experiments revealed that: (i) only a subset of the genes bound by the floral homeotic factors respond to a perturbation of their activities; and (ii) despite their similar genome-wide binding patterns (see above), the different floral organ identity factors regulate markedly distinct sets of target genes [9,10,25,34]. This discrepancy between the data from genome-wide localization and perturbation experiments suggests that determinants for the regulatory specificity of the floral organ identity factors other than their inherent DNA-binding activities must exist. In fact, models for the control of gene expression in animals suggest that transcription factors often act in combination with other transcriptional regulators in higher-order protein complexes [35,36]. The assembly of these complexes is thought to be highly dynamic, involving transcription factor collectives that cooperatively control gene expression, thus supplying target gene response specificity [37,38]. Notably, it has been shown that one of the founding members of the MADS-domain family, SERUM RESPONSE FACTOR (SRF) from humans (see above), interacts with cofactors belonging to the ternary complex factor and the myocardin-related transcription factor families, respectively, and that the regulation of these cofactors by different signaling pathways affects the control of SRF target genes [39,40]. If such models also applied to plants, the specificities of the different floral organ identity complexes could be in part a result of their interactions with different combinations of additional transcription factors on the promoters of target genes. In other words, binding of a floral organ identity factor complex to a CArG-box sequence (or a pair of CArG-boxes in the case of a MADS-domain protein tetramer) in a given promoter might only mark a gene as a potential target, with a transcriptional response then depending on: (a) whether the promoter also contains binding sites for additional regulators; and (b) whether the different transcription factors bound to the promoter can interact with each other and form a transcription factor collective. Thus, the regulatory output would depend on the use of the proper "motif grammar”, i.e., the identity, number, and exact positioning of binding sites in a promoter [41]. 

In support of such a scenario, several lines of evidence suggest that the floral organ identity factors are part of larger transcriptional complexes: (i) Gel filtration experiments showed that protein complexes containing SEP3 have a molecular weight of around 670 kDa, which is considerably larger than MADS-domain protein tetramers (~120 kDa) [19]. (ii) There is an over-representation of sequence motifs that are known binding sites of non-MADS-domain transcription factors in the vicinity of CArG-boxes [9,22]. These include G-boxes, which are bound by transcription factors of the basic helix-loop-helix (bHLH) and basic leucine zipper (bZIP) families, and binding sites for class I and class II TCP transcription factors. (iii) Immunoprecipitation experiments coupled to mass spectrometry (IP-MS) [19], high-throughput yeast two-hybrid assays [42], and other experimental approaches yielded several candidates for proteins interacting with the floral organ identity factors. These include transcription factors of different families (Table 1), of which some (e.g., members of the TCP family) may be able to bind to the non-CArG-box motifs that are over-represented in target gene promoters. However, in most cases, the biological significance of these predicted interactions remains to be elucidated. One notable exception comes from a study showing that BASIC PENTACYSTEINE (BPC) transcription factors facilitate the binding of a transcriptional repressor complex containing AP1 to the promoter of the ovule identity gene *SEEDSTICK* [43]. Despite this progress, a clear picture of how cofactors interact with and confer specificity on the different floral organ identity factor complexes has yet to emerge. 

In addition to transcription factors, the above-mentioned experimental approaches also yielded epigenetic regulators as candidate interactors (Table 2). For example, the chromatin remodeler CHROMATIN REMODELING 4 (CHR4) was identified in IP-MS experiments with different floral organ identity factors [19]. Notably, these interactions were confirmed in a reciprocal experiment where CHR4 was immunoprecipitated [50]. Furthermore, it was shown that CHR4 controls the deposition of certain histone marks at selected genomic loci and that its activity affects the expression of several floral regulatory genes [50]. Thus, CHR4 and other epigenetic regulators likely mediate the regulatory output of the floral organ identity factor complexes. Because the floral MADS-domain proteins have been shown to be bi-functional transcription factors [9,10,25,26,27,28], it appears likely that they interact with either transcriptional activators or repressors depending on the genomic context. However, how the composition of the complexes is determined to ensure the proper regulatory output is currently unknown.

In summary, several independent lines of evidence support the idea that the floral organ identity factors are part of large regulatory complexes that contain cofactors and epigenetic regulators (Figure 2). The exact composition of these complexes is currently unknown, and how the different complex components contribute to conferring functional specificity is not well understood.

## 4. Other Possible Determinants for Specificity

There is ample evidence for the role of post-translational modifications (PTMs) in the regulation of transcription factor activities [53]. Common PTMs include phosphorylation, acetylation, methylation, sumoylation, and ubiquitination, which can affect virtually all aspects of a transcription factor’s function by controlling their subcellular localization, stability, DNA-binding activity, and regulatory specificity [53]. To date, very little is known about the extent to which the floral organ identity factors are post-translationally modified. In one study, several floral organ identity factors were identified as possible substrates of mitogen-activated protein (MAP) kinases [54]. Notably, phosphorylation of the MADS-domain of human SRF and Myocyte Enhancer Binding Factor-2 (MEF2) has been demonstrated and has been suggested to regulate the DNA-binding activities of these proteins [55,56]. Additionally, the Arabidopsis MADS-domain protein AGAMOUS-LIKE15 (AGL15) was shown to be phosphorylated by MAP kinase kinases in flowers, likely increasing its activity [57]. Thus, there is a clear precedent for the regulation of MADS-domain transcription factors by phosphorylation. However, whether the floral organ identity factors are phosphorylated in vivo is currently unknown.

One published example of a PTM of a floral organ identity factor comes from a study that demonstrated farnesylation (a form of prenylation) of AP1 [58]. It has been suggested that this modification may regulate the specificity of AP1 by affecting its interactions with other proteins. However, it was later shown that mutating the farnesylation motif in AP1 does not significantly alter AP1 activity [59], and thus the role of this specific PTM remains enigmatic.

Another mechanism that is well known to affect transcription factor function is alternative splicing, where certain protein domains are either retained or eliminated based on the composition of the mature mRNA after splicing of the corresponding pre-mRNAs. While alternative splicing is not specific to transcription factor-coding genes and likely affects over 60% of intron-containing genes in higher plants [60], transcription factors may be particularly susceptible to the effects of alternative splicing, especially if they interact with other regulatory proteins [61]. Specifically, the activity of a given regulator might be dependent on the isoforms of multiple proteins that interact as part of a transcription factor collective. Notably, it has been shown that alternative splicing occurs frequently in transcripts of plant MADS-box genes, including some of the floral homeotic genes [62]. These splicing events primarily affect domains in the carboxy-terminal half of the transcription factors that are known to mediate protein-protein interactions, including the formation of dimers and tetramers and likely the assembly of larger regulatory complexes as well. Because protein-protein interactions are pivotal for the functions of the floral organ identity factors (see above), alternative splicing may be essential for regulating their activities. The dramatic effects that alternative splicing can have on the activities of MADS-domain transcription factors were demonstrated for FLOWERING LOCUS M (FLM), which is involved in the temperature-dependent control of flowering time in Arabidopsis. It was shown that *FLM* transcripts undergo alternative splicing and that different ambient temperatures lead to the predominant accumulation of one or another splice form [63,64]. The proteins synthesized from these alternative transcripts interact with another MADS-domain protein, SHORT VEGETATIVE PHASE (SVP), resulting in complexes that either promote or repress flowering depending on the FLM isoform incorporated into the dimer.

## 5. Lessons from *Hox* genes

As described above, the activities of transcription factors are known to be controlled by a variety of mechanisms. To better understand which of these mechanisms contribute to the regulation of the floral organ identity factors, it may be instructive to review what is known about functionally analogous transcription factors in animals. We believe that Hox proteins are particularly interesting in this context because they play roles similar to those of the floral organ identity factors, in that they control the specification of different organ and cell types along an axis [65]. Their inactivation often leads to organ transformations [66], exactly as observed for the floral homeotic mutants. Another parallel between the floral homeotic and *Hox* genes is that they are typically expressed only in the cells, tissues, or organs they help to specify [65]. Furthermore, the regions in which different *Hox* genes are expressed often overlap, so that in some cells combinations of Hox proteins are active [65], just as was observed for the floral homeotic genes. Yet despite the functional similarities between these classes of master regulators, *Hox* genes do not encode MADS-domain but homeodomain transcription factors [65,67,68]. These regulatory genes must therefore have evolved independently. Due to the nonhomologous origin of *Hox* and floral organ identity genes, any comparison between the regulation of the corresponding transcription factors has to be approached with caution; however, given the striking similarities in their functions and the fact that evolutionary mechanisms are often conserved between plants and animals [69], it may nevertheless be informative. In fact, even at the molecular level, there are parallels: similar to MADS-domain transcription factors, different Hox proteins bind to specific DNA sequence motifs that are frequently found in animal genomes [65,67,68]. Despite all Hox proteins binding to similar sequences, differences in their in vitro and in vivo binding site preferences as well as in their binding affinities have been detected [65,67,68]. Based on the detailed analysis of promoters targeted by Hox proteins, different classes of target genes have been defined largely depending on the degree of binding site specificity—some sites appear to be bound by only one Hox protein, while others are non-specific and can be targeted by many different Hox factors [65,67,68]. Furthermore, Hox proteins may bind to low affinity sites and even to sites that do not contain their consensus binding motif. It has been suggested that binding to these sites can be efficiently regulated by controlling their accessibility in chromatin [70,71].

In addition to these variations in binding specificities, the regulation of Hox target genes appears to be dependent in some cases on the presence of cofactors. Arguably the best studied of the Hox cofactors are members of the Three Amino Acid Loop Extension (TALE) class of homeodomain-containing proteins [65,67,68]. These proteins not only function as Hox cofactors but also have other regulatory functions during development [65], making their characterization through genetic approaches difficult. It is thought that the interaction of Hox and TALE proteins can both activate and repress gene expression depending on the combinations of the interacting proteins [67]. Furthermore, it was shown that the binding specificities of Hox proteins can change when they interact with TALE proteins [72]. In addition to these cofactors, proteins have been identified that are recruited to *cis*-regulatory modules bound by the Hox transcription factors but that may not bind DNA cooperatively with them [65,67,68]. These proteins are thought to help with the assembly of regulatory complexes and/or mediate part of the regulatory output. In summary, it appears that the regulatory specificity of Hox proteins is determined by a complex interplay of different mechanisms [73], with the exact mode of action depending not only on the properties of a particular Hox protein but also on the nature of the individual target genes.

## 6. Conclusions

In this review, we have summarized some of the current knowledge about the function and regulation of the floral organ identity factors. While insights into the role of dimerization and tetramerization as well as differences in DNA-binding preferences have been obtained, other regulatory mechanisms that could be at play have not yet been studied in detail. In particular, it seems important to better characterize the composition and dynamics of the regulatory complexes that contain the floral organ identity factors and to study the functions of the different complex components. The identification of several candidates for interactors in recent years has provided a good starting point for this work. Based on what has been found for the TALE proteins interacting with Hox transcription factors in animals (see above), we think it likely that cofactors of the floral MADS-domain proteins also have other developmental functions and may not even be specifically expressed in flowers. This, and the high level of functional redundancy often found among plant genes, could explain why such cofactors have not been identified in the many genetic screens that were conducted in Arabidopsis. It is also possible that there are many cofactors from different families that each control only a small subset of target genes. Thus, there may not be one general mechanism underlying floral organ identity factor function but rather a combination of different and possibly interacting mechanisms that may be specific for each regulatory complex and that might vary depending on the genes they target. With the ever-improving experimental approaches plant biologists have at their disposal, now seems to be a good time to reach to the bottom of how the floral organ identity factors achieve regulatory specificity.

## Figures and Tables

**Figure 1 plants-12-01128-f001:**
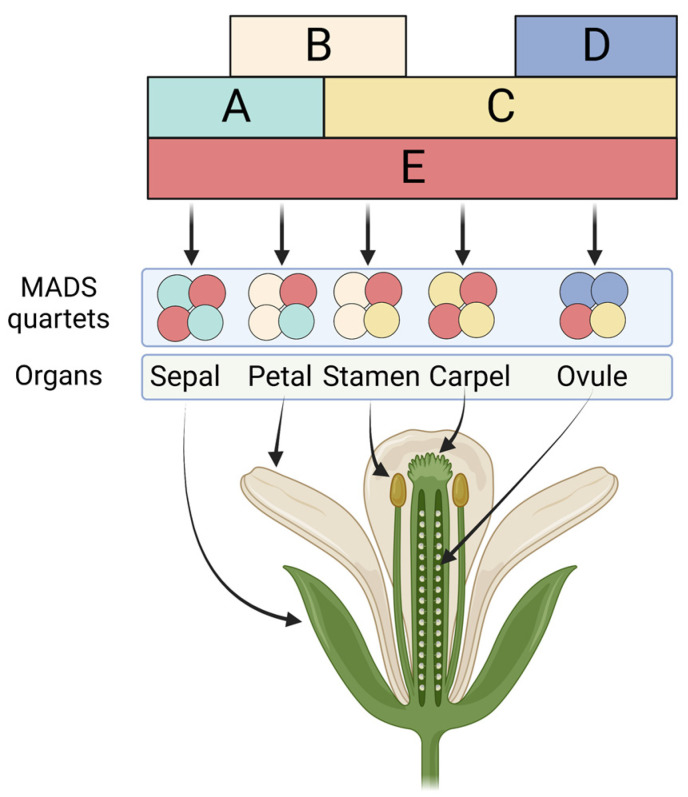
The ABCDE model of floral organ identity specification. The identity of the different floral organs is specified by the combinatorial activity of A-, B-, C-, D-, and E-class genes (as indicated). The MADS-domain transcription factors encoded by these genes act together in different tetrameric complexes (‘quartets’) to control the developmental programs needed for the formation of sepals, petals, stamens, carpels, and ovules. Colors indicate the composition of the different MADS-domain protein quartets. Figure created with BioRender.com.

**Figure 2 plants-12-01128-f002:**
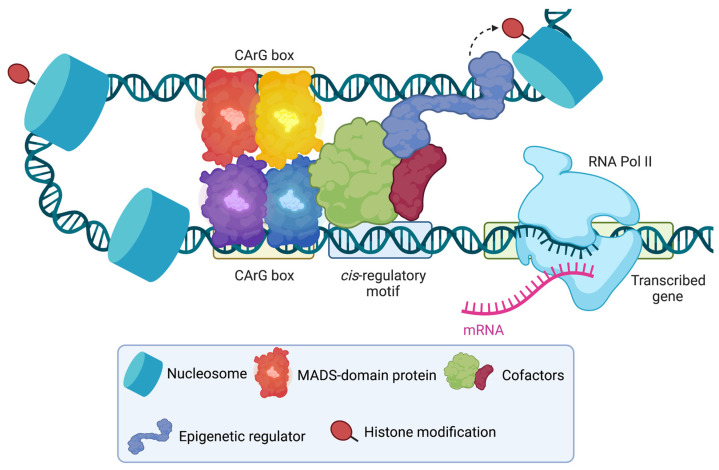
Possible mechanisms for floral organ identity factor function. Two dimers of MADS-domain proteins bind to adjacent CArG boxes. The formation of a tetrameric complex as a result of the interaction of the two dimers leads to looping of the interjacent DNA [16,52]. Cofactors bind to *cis*-regulatory elements in the vicinity of CArG-boxes and interact with the floral organ identity factors, the transcriptional machinery, and/or recruit epigenetic regulators, which can alter chromatin structure and accessibility, e.g., by making post-translational modifications to histones. Figure created with BioRender.com.

**Table 1 plants-12-01128-t001:** Non-MADS-domain transcription factors identified as interactors of floral organ identity factors.

	Protein Alias	Family	Interacting with	Evidence	Reference
AT1G02065	SPL8	SQUAMOSA-promoter binding protein	AG, AP1	IP-MS	[19]
AT1G08970	NF-YC9	NF-Y	AG	Two-hybrid	[42]
AT1G13400	NUB	Zinc finger	AG	Two-hybrid	[44]
AT1G19220	ARF19	Auxin Response Factor	SEP2	Two-hybrid	[42]
AT1G63910	MYB103	MYB	AG	Two-hybrid	[42]
AT1G68240	-	Basic helix-loop-helix	SEP2	Two-hybrid	[42]
AT1G69690	TCP15	TCP	AG, AP1, PI, SEP4	Two-hybrid	[42,44,45]
AT1G77450	NAC032	NAC domain containing protein	SEP2	Two-hybrid	[42]
AT2G01930	BPC1	Basic pentacysteine	AP1	Two-hybrid	[43]
AT2G17950	WUS	Homeodomain	SEP3	Two-hybrid	[44]
AT2G35940	BLH1	Homeodomain	AP1	IP-MS	[19]
AT2G37630	AS1	MYB	AG	Two-hybrid	[44]
AT3G11100	VFP3	SANT and trihelix	SEP4	Two-hybrid	[42]
AT3G15030	TCP4	TCP	AP1, SEP4	Two-hybrid	[42]
AT3G19070	-	Homeodomain-like	SEP1	Two-hybrid	[42]
AT3G24490	-	MYB/SANT-like	SEP1	Two-hybrid	[42]
AT3G46600	-	GRAS	AG, SEP1	Two-hybrid	[42]
AT3G47620	TCP14	TCP	AP1, SEP4	Two-hybrid	[42]
AT3G51080	GATA6	GATA	PI	Two-hybrid	[42]
AT4G01580	-	AP2/B3-like	PI	Two-hybrid	[42]
AT4G03250	-	Homeodomain-like	SEP1	Two-hybrid	[42]
AT4G14225	-	Zinc finger	PI	Two-hybrid	[42]
AT4G15250	BBX9	Zinc finger	AG	Two-hybrid	[42]
AT4G18390	TCP2	TCP	AG	Two-hybrid	[42]
AT4G37740	GRF2	Growth regulating factor	PI	Two-hybrid	[42]
AT5G02030	RPL	Homeodomain	AP1	IP-MS	[19]
AT5G05120	-	Zinc finger	PI	Two-hybrid	[42]
AT5G09750	HEC3	Basic helix-loop-helix	PI	Two-hybrid	[42]
AT5G11270	OCP3	Homeodomain	SEP1	Two-hybrid	[45]
AT5G24470	PRR5	Pseudo-response regulator	PI	Two-hybrid	[42]
AT5G38480	GRF3	Growth regulating factor	PI	IP-MS	[46]
AT5G41410	BEL1	Homeodomain	AG, SEP3	Reconstituted Complex	[47]
AT5G41920	SCL23	GRAS	SEP4	Two-hybrid	[48]
AT5G61850	LFY	FLO/LFY	SEP3	Reconstituted Complex	[49]

Identifiers of the corresponding genes, protein aliases, and transcription factor families to which the different interactors belong are shown. Information on the experimental methods used for the identification of interactors (‘Evidence’) and the relevant references are provided.

**Table 2 plants-12-01128-t002:** Epigenetic regulators identified as interactors of floral organ identity factors.

Gene ID	Protein	Family	Interacting with	Evidence	Reference
AT5G44800	CHR4	CHD3	AG, AP1, PI, SEP3	IP-MS	[19,50]
AT3G06400	CHR11	ISWI	AG, AP1, AP3, PI, SEP3	IP-MS	[19]
AT4G39100	SHL1	PHD finger, BAH-domain	AP3	Two-hybrid	[42]
AT1G43850	SEU	adn1/SEU	AP1, AP3, SEP3	Two-hybrid; Reconstituted Complex, IP-MS	[19,51]
AT5G18620	CHR17	ISWI	AG, AP1, PI, SEP3	IP-MS	[19]
AT3G48430	REF6	JHDM3 histone demethylase	AG, AP1, SEP3	IP-MS	[19]
AT2G25170	PKL	CHD3	SEP3	IP-MS	[19]
AT2G32700	LUH	WD40/ LUFS domain	AP1, SEP3	IP-MS	[19]

Identifiers, protein aliases, and families to which the different interactors belong are shown. Information on the experimental methods used for the identification of interactors (‘Evidence’) and the relevant references are provided.

## Data Availability

Not applicable.

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
