# Peer review of "Floral Homeotic Factors: A Question of Specificity"

_plants, 2023, doi:10.3390/plants12051128_

Round 1

Reviewer 1 Report

Authors present an interesting review about MADS-factors implicated in determining floral organ identity. In general the text is well written and important nowadays questions are well described. However I have a few concerns:

In my personal opinion, the title do not reflect the real content of the manuscript. The title suggest a more wider topic, but the review only talk about MADS transcription factors. I think title can be more specific and it do not affect the interest in the topic.

Second, I have found intriguing why authors have talk about ABCE model, when it seems they talk about plants in general, but do not say anything about the ABCDE model.  I know D genes are not present in all plants, but I think the model must be presented complete, at least given the complete description.

When describing DNA binding of MADS TF, authors talk about motifs (CArG-boxes), but it is nowadays clear that matrices describes better the binding properties of TFs. Actually, the classical "motif boxes" usually represents the more restrictive sites in the matrix and usually only a part of the complete DNA region that interacts with TFs. In fact, some DNA sequences cannot match the "motif box" but have a good binding affinity. This is specially relevant for the comments in lines 294-296, but also in other part in the manuscript, as we already know since more than a decade, that some functionally bound sites in the genome do not match the "consensus" motif and it is not necessary to find explanations in other cofactors as in many cases it's explained when an affinity matrix is considered. Of course, in some cases, even the matrix did not explain the binding and interaction with other factos can explain it.

I have the feeling, that even if comments about Hox genes are interesting, in comparison with the total extension of the manuscript the section is too long. Actually, some of the similarities between MADS and Hox do not give many information as it is not better know in animals than in plants.

Author Response

Point 1: “In my personal opinion, the title do not reflect the real content of the manuscript. The title suggest a more wider topic, but the review only talk about MADS transcription factors. I think title can be more specific and it do not affect the interest in the topic.”

Response: We have considered this suggestion and changed the title slightly to hopefully avoid confusion as to the content of the review.

Point 2: “Second, I have found intriguing why authors have talk about ABCE model, when it seems they talk about plants in general, but do not say anything about the ABCDE model.  I know D genes are not present in all plants, but I think the model must be presented complete, at least given the complete description.”

Response: We now briefly mention D class genes (lines 36-38), as requested. We also show the ABCDE model in a new Figure 1.

Point 3: “When describing DNA binding of MADS TF, authors talk about motifs (CArG-boxes), but it is nowadays clear that matrices describes better the binding properties of TFs. Actually, the classical "motif boxes" usually represents the more restrictive sites in the matrix and usually only a part of the complete DNA region that interacts with TFs. In fact, some DNA sequences cannot match the "motif box" but have a good binding affinity. This is specially relevant for the comments in lines 294-296, but also in other part in the manuscript, as we already know since more than a decade, that some functionally bound sites in the genome do not match the "consensus" motif and it is not necessary to find explanations in other cofactors as in many cases it's explained when an affinity matrix is considered. Of course, in some cases, even the matrix did not explain the binding and interaction with other factos can explain it.”

Response: We fully agree with the reviewer’s comments and while we do not mention matrices here specifically, we do stress that variations in binding site sequences contribute substantially to conferring specificity. To specifically address this comment, we included a sentence (lines 97-100) to say that not all known binding sites contain CArG-box sequences, suggesting that in some cases, the floral homoetic factors can bind to other, potentially unrelated motifs.

Point 4: “I have the feeling, that even if comments about Hox genes are interesting, in comparison with the total extension of the manuscript the section is too long. Actually, some of the similarities between MADS and Hox do not give many information as it is not better know in animals than in plants.”

Response: We have included the section on Hox proteins not only because of their functions being similar to those of the floral homeotic factors, but also because in recent years, considerable progress has been made in understanding the mechanisms regulating their specificity. However, we do agree with the reviewer’s comment that this section is too long and accordingly, have shortened it.

Reviewer 2 Report

I found that this is a timely, interesting and well-written review.

I do not have any major concerns.

 Minor points:

I miss the reference of Lee et al. 2013 (Science) about FLM/SVP (lines 250-260). 

Figure 1: some nucleosome cartoons show a a tag, is it a post-translational modification? please clarify in the figure or the legend.

Author Response

Point 1: “I miss the reference of Lee et al. 2013 (Science) about FLM/SVP (lines 250-260). 

Figure 1: some nucleosome cartoons show a a tag, is it a post-translational modification? please clarify in the figure or the legend.”

Response: We have added the missing reference and clarified the use of the tag in Figure 1.

Reviewer 3 Report

I Have finished now the revision of the review with title "Floral Organ Identity Factors: A Question of Specificity" and found it acceptable for publication with only a few changes most of the them suggestions relative to the style and/or typos.

I attach the pdf file for the authors to consider the changes proposed.

Best regards

Author Response

Response: reviewer 3 provided comments directly in the manuscript file and suggested a number of minor text changes, which we have addressed.

Reviewer 4 Report

The review address recent progress in the functional specificity of the floral organ identity factors, an interesting topic for the scientific community. I found it quite well written with pertinent subheads and addressing the question that it promises with the title. However, I still have some concerns and suggestions, the most important ones related to the animal part:

1.- As they are explaining the ABCE model, I suggest to add one figure representing it with the important factors that play a role in it (could be an adaptation from another paper), or add more explanation in the introduction, presenting all the genes that belong to that model,... Maybe not all the audience know all the genes that are participating in that model and how they work.

2.- Fig. 1 is good but too simple and not giving too much information. I suggest to improve it, maybe including somewhere the epigenetic regulators and how they affect the expression.

3.- Related to the epigenetic regulators, the BPC-AP1 complex could be also repressing STK by recruiting proteins from the PRC2 complex, as it was reported previously for other genes. The authors could highlight this and using it as a clear link between cofactors and epigenetic regulators (maybe this gives to them an idea to put cofactors, and epigenetic regulators in the same figure).

4.- Finally and regarding the animal part, I found quite interesting that the authors jump to animals in order to give more insights about how other transcription factors better characterized in terms of specificity works and then try to think mechanisms that could influence the floral organ identity genes. However, from my point of view, it is not adding too much interesting information. First of all, Hox protein are really different from MADS-box protein making difficult to extrapolate the results to the floral organ identity genes, as the authors said. And second, Hox genes are doing more or less things that are already reported with MADS-box in plants. For example, TALE-Hox complex specificity is more or less similar as the one explained in Balanza et al (2014) J. exp. bot, although in that case the complex is made always by two MADS-box (not a cofactor). In addition, in that paper explain that in some cases one of the MADS-box is participating in the transcriptional activation of the genes but without interacting with the DNA, as the example present in line 309.

From my point of view, it would be more useful to arrange this part focusing on the SRF transcriptions factors from human, the ones that are more similar to MADS-box of plants. In fact, they already mention it during the text some examples (line 145-148 and 222-224). Thus, I suggest to change the title of this subhead in order to add all the information collected from animal and focus more on the SRF. 

Author Response

Point 1: “As they are explaining the ABCE model, I suggest to add one figure representing it with the important factors that play a role in it (could be an adaptation from another paper), or add more explanation in the introduction, presenting all the genes that belong to that model,... Maybe not all the audience know all the genes that are participating in that model and how they work.” 

Response: We initially decided against including such a figure because we felt that this would be rather off-putting for experts who may view our manuscript as just another review on the ABCDE model. But we do get the reviewer’s point that this is important for the non-expert audience and accordingly, have included it as a new Figure 1.

Point 2: “Fig. 1 is good but too simple and not giving too much information. I suggest to improve it, maybe including somewhere the epigenetic regulators and how they affect the expression.” 

Response: As requested, we have modified the figure to highlight the involvement of epigenetic regulators.

Point 3: “Related to the epigenetic regulators, the BPC-AP1 complex could be also repressing STK by recruiting proteins from the PRC2 complex, as it was reported previously for other genes. The authors could highlight this and using it as a clear link between cofactors and epigenetic regulators (maybe this gives to them an idea to put cofactors, and epigenetic regulators in the same figure).”

Response: This is a good point. As described above, we have modified the figure to highlight the activities of epigenetic regulators.

Point 4: “Finally and regarding the animal part, I found quite interesting that the authors jump to animals in order to give more insights about how other transcription factors better characterized in terms of specificity works and then try to think mechanisms that could influence the floral organ identity genes. However, from my point of view, it is not adding too much interesting information. First of all, Hox protein are really different from MADS-box protein making difficult to extrapolate the results to the floral organ identity genes, as the authors said. And second, Hox genes are doing more or less things that are already reported with MADS-box in plants. For example, TALE-Hox complex specificity is more or less similar as the one explained in Balanza et al (2014) J. exp. bot, although in that case the complex is made always by two MADS-box (not a cofactor). In addition, in that paper explain that in some cases one of the MADS-box is participating in the transcriptional activation of the genes but without interacting with the DNA, as the example present in line 309. 

From my point of view, it would be more useful to arrange this part focusing on the SRF transcriptions factors from human, the ones that are more similar to MADS-box of plants. In fact, they already mention it during the text some examples (line 145-148 and 222-224). Thus, I suggest to change the title of this subhead in order to add all the information collected from animal and focus more on the SRF.“

Response: We agree with the reviewer’s comment that one has to be cautious when comparing Hox proteins to the floral homeotic factors and we specifically acknowledge this in the manuscript. We also agree (and say so in the manuscript) that there are many known parellels between these master regulators. This is in fact one of the reasons why we wrote this section. However, we feel that the Hox protein field is more advanced when it comes to the understanding of how acombination of different mechanisms contributes to conferring specificity. In this section, we want to highlight and emphasize this idea/concept.

Regarding SRF: we had already included important information about its regulation, as acknowledged by the reviewer. We decided against dedicating a full section to it and rather to focus on Hox proteins as they are functionally analogous and also act in a combinatorial manner.

In order to stimulate thought and discussion, we would like to keep the structure as it is. To directly address the reviewer’s comments, we have shortened the section on Hox genes to put less weight on it.